# THE VEP BOOSTER: A CLOSED-LOOP AI SYSTEM FOR VISUAL EEG BIOMARKER AUTO-GENERATION

## ABSTRACT

The effectiveness of Visual Brain-Machine Interfaces (BMIs) is significantly dependent on the accurate detection and interpretation of electroencephalography (EEG) biomarkers, which frequently exhibit variability due to physiological changes and environmental disturbances over time. Traditional EEG signal enhancement strategies largely concentrate on signal processing techniques such as feature extraction and filtering; however, these approaches often do not adequately address the inherent sources of variability that affect biomarker stability over time. To surmount these challenges, we have developed the Visual Evoked Potential Booster (VEP Booster), a novel closed-loop artificial intelligence framework designed to produce reliable and stable EEG biomarkers under visual stimulation protocols. Our system utilizes a Deep Convolutional Generative Adversarial Network (DCGAN) to refine stimulus images based on real-time feedback from human EEG signals, thereby creating visual stimuli that are specifically tailored to the characteristic preferences of neurons in the primary visual cortex. We evaluated the efficacy of this system through the implementation of steady-state visual evoked potential (SSVEP) protocols in nine human subjects. In our evaluations, both the SSVEP biomarker amplitude and the single-trial SSVEP binary classification experiments, encompassing intra- and inter-temporal analyses, exhibited statistically significant enhancements when employing the VEP Booster. These encouraging outcomes underscore the potential for broad applications in clinical and technological domains.

## 1 INTRODUCTION

Visual Brain-Machine Interfaces (BMIs) have emerged as key technologies that bridge neural activity with external devices, enabling transformative applications in neurorehabilitation (Astrand et al., 2014), diagnosis of brain disorders (Zhang et al., 2024), and human-computer interaction (Chen et al., 2015; Zhu et al., 2022). The performance of visual BMIs critically depends on the detection and interpretation of reliable and stable electroencephalography (EEG) biomarkers. However, in real-life scenarios, EEG biomarkers often exhibit significant variability over time, which poses a major challenge to consistent BMI performance (Lotte et al., 2018).

The instability of EEG biomarkers is mainly due to several factors (Buzsáki et al., 2012), including minute differences in electrode placement, amplitudes attenuations due to change in physiological and psychological states of subjects, environmental noise and individual differences. These factors lead to significant variations in the EEG signals at different time points.

Despite significant advancements, most current EEG signal enhancement algorithms remain focused on signal processing techniques such as filtering (Sanei & Chambers, 2013; Agarwal et al., 2017; He et al., 2004) denoising (Luck, 2014; Makeig et al., 1996). Additionally, feature extraction methods are applied to transform the enhanced signals into meaningful representations. While these methods can effectively reduce noise and emphasize certain features, they often overlook the inherent temporal dynamics of EEG biomarkers. This oversight means they cannot fundamentally address the issue of biomarker instability over time. As a result, models trained earlier data may experience performance degradation when applied at later times, a challenge that remains urgent to resolve.

In this work, we introduce the *Visual Evoked Potential Booster* (VEP Booster), a novel closed-loop artificial intelligence framework that fundamentally differs from existing methodologies. Our

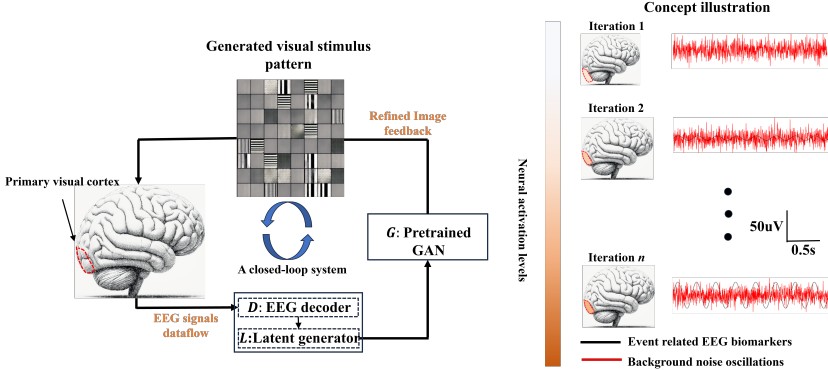

Figure 1: Overview of the VEP Booster from a closed-loop perspective. The system comprises a pre-trained Generative Adversarial Network (GAN), a latent vector generator, and an EEG decoder, as demonstrated on the left-hand side of the figure. As the system interacts with a participant, event-related EEG biomarkers are significantly enhanced, while background noise oscillation is reduced, as demonstrated on the right-hand side of the figure.

approach signifies a conceptual breakthrough by shifting from passive signal processing to active EEG biomarker sources (Bashivan et al., 2019; Ponce et al., 2019) through stimulus optimization. Instead of filtering or decoding EEG signals *post hoc*, the essence of a VEP booster lies in identifying the visual stimulus feature preferences of neurons under a specific visual paradigm. It capitalizes on the intrinsic preferences of specific neurons to generate optimal stimuli, thereby enhancing the activations of neurons that generate reliable EEG biomarkers.

As shown in Figure 1, our approach leverages a Deep Convolutional Generative Adversarial Network (DCGAN) within a closed-loop system to generate and refine visual stimuli based on real-time EEG feedback. As a result, the amplitude of the EEG biomarkers is increased, which improves their resistance to the effects of background oscillations in the brain across different time periods.

We validated the efficacy of the VEP Booster through a series of experiments employing SSVEP protocols with nine human participants. The results demonstrated a significant average increase of 106% in SSVEP biomarker values across individuals (ranging from 28% to 295%). For algorithm evaluations, in single-trial cross-time experiments conducted between 10AM and 10PM, the VEP Booster statistically improved the SSVEP binary classification accuracy of the Compact Convolutional Neural Network (CCNN) (Ravi et al., 2020), EEGNet (Lawhern et al., 2018) and SSVEPNet models (Pan et al., 2022a).Our key contributions are as follows:

• **Algorithm**: We develop and validate a novel closed-loop AI framework that actively generates reliable and stable EEG biomarkers through optimized visual stimuli, addressing major challenges of visual EEG biomarker signal variations across time.

• **Applications**: To demonstrate the feasibility of our proposed framework, we have developed a comprehensive SSVEP EEG biomarker enhancement system. By improving the stability and reliability of SSVEP biomarkers, our system lays a solid foundation for practical applications and has the potential to make a profound impact on technological applications such as human-computer interaction.

• **Dataset**: We will open-source the cross-time SSVEP EEG dataset (containing 4500 data samples collected from three participants), contributing valuable resources to the community for exploring visual brain-machine interfaces.

## 2 RELATED WORK

Existing visual BMI EEG processing algorithms can generally be divided into three categories. The first category relies on traditional signal processing techniques. Methods such as Canonical Correlation Analysis (CCA) (Zhang et al., 2011) and Task-Discriminant Component Analysis (TDCA) (Liu et al., 2021) have achieved significant success in detecting steady-state visual evoked potentials

(SSVEPs). In a recent study, the binocular Task-Related Component Analysis (bTRCA) algorithm in visual BMIs achieved impressive information transmission rates (Sun et al., 2024).

The second category includes machine learning approaches utilizing deep neural networks (DNNs) to extract EEG biomarkers. DNNs are particularly effective at capturing spatial and temporal dependencies in EEG data (Pan et al., 2022b; Liu et al., 2023). For instance, Kwak et al. (2017) demonstrated that convolutional neural networks (CNNs) outperform traditional methods like CCA and its variants, achieving high classification accuracy in both static and ambulatory settings. Recently, the application of large language models and transformer architectures to EEG data (Yi et al., 2024; Jiang et al., 2024; Duan et al., 2023)is promising and exciting. Pre-trained transformer models offer a robust framework for interpreting EEG signals by leveraging universal patterns across various tasks.

The third category Spiking Neural Networks (SNNs), utilize surrogate-gradient descent learning to reconstruct EEG biomarker sources and show potential for augmenting EEG signal data (Singanamalla & Lin, 2021). The results indicate improvements in tasks such as Steady-State Visual Evoked Potentials (SSVEP) and motor imagery. Additionally, biologically-motivated Spiking Recurrent Neural Networks (SRNNs) employ the FORCE method to train network dynamics to align with EEG dynamics (Ioannides et al., 2022).

## 3 THE VEP BOOSTER

### 3.1 MATHEMATICAL MODEL

The primary objective of the VEP booster is to generate images that elicit the strongest EEG biomarker responses under a visual stimulus protocol. For a single loop, the system works as follows: 1) a generator generates visual stimulus images; 2) we record the human participant's EEG signals under each visual stimulus protocol with the generated images; 3) an EEG decoder scores the generated images based on the EEG biomarker values; 4) a latent generator generates the latent inputs (for the DCGAN) for the next iteration, which based on the latent vectors of images with highest EEG biomarker values.

As illustrated in Figure 1, the system comprises of four interconnected sub-systems:

• 1) **Generator**, denoted as $G$; Without loss of generality, we use a pre-trained Generative Adversarial Network (GAN) to implement the generative model $G : \mathcal{Z} \to \mathcal{X}$, where $\mathcal{Z}$ is the latent space, and $\mathcal{X}$ is the space of visual stimuli. $G$ is defined by parameters $\theta_g$, with the optimization objective to minimize the discrepancy between the generated visual stimuli and the dataset visual stimuli.

• 2) **EEG Decoder**, denoted as $D$; Formally, $D$ is the mapping from EEG signals to scores, $D : \mathcal{E} \to \mathbb{R}$, outputting a real-number score indicating the effectiveness of the stimulus. $D$ is defined by parameters $\theta_d$, with the goal to accurately predict the quality and the differences of the visual stimuli corresponding to the given EEG signals.

• 3) **Latent Generator**, denoted as $L$; $L$ be a function that modifies the latent vectors $z$ from the distribution $p(z)$ based on the scores derived from the EEG decoder, formulated by:

$$z_{\text{mutated}} = \arg \max_z \sum_{i=1}^{n} \text{feature}_i(z) + \delta, \tag{1}$$

where $\text{feature}_i(z)$ are functions returning the various feature scores of the EEG signals associated with latent vector $z$, and $\delta$ is a mutation vector where each element $\delta_i$ is drawn from a normal distribution $\mathcal{N}(0, \sigma^2)$ with probability $p$, representing the mutation rate. The variance $\sigma^2$ controls the extent of mutation. The number of feature vectors depends on visual protocols.

• 4) **Human Subject**, who actively participates in the closed-loop process. We define an EEG Signal Model $f$, which is the mapping from visual stimuli to EEG signals from the human brain, $f : \mathcal{X} \to \mathcal{E}$, where $\mathcal{E}$ represents the EEG signal space. This mapping is typically constrained by the neurophysiological processes of the brain and can be highly nonlinear and complex.

The above four sub-systems operate in a closed-loop manner to optimize the following goal:

$$\max_{\theta_g} \mathbb{E}_{z \sim p(z)}[D(f(G(L(z, \alpha); \theta_g); \theta_d))], \tag{2}$$

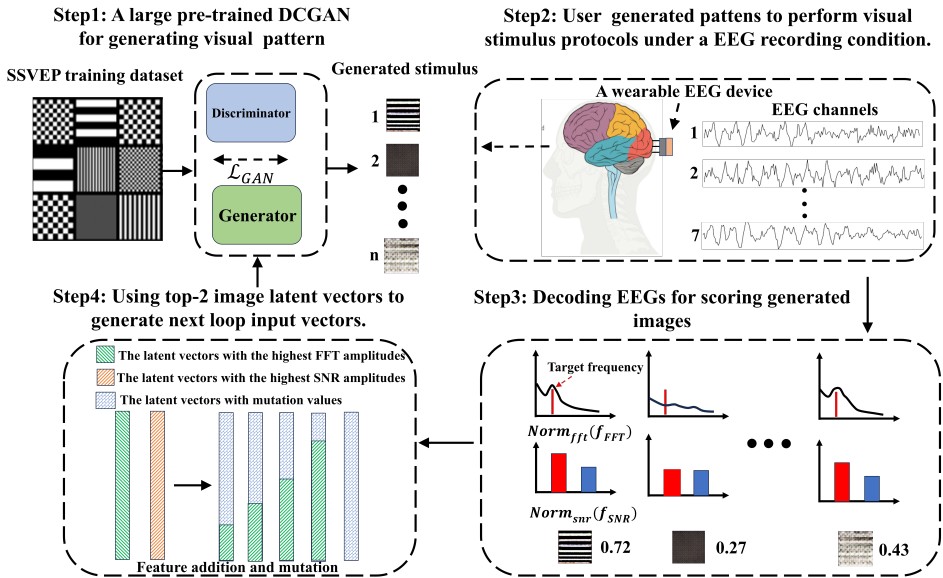

Figure 2: The system implementation. The process involves four main steps: (1) Utilization of a pre-trained DCGAN to create visual stimuli; (2) Presentation of these stimuli to subjects during EEG recording sessions, following Steady State Visual Evoked Potential (SSVEP) protocols; (3) Decoding and analysis of the EEG data to evaluate the efficacy of the generated images; (4) Selection and use of the top two performing image latent vectors from the current iteration to refine and generate input vectors for the subsequent iteration.

where $p(z)$ is the prior distribution of the latent space. This goal facilitates the development of a GAN capable of generating visual neuron preferred stimulus from real data by continuously modifying latent vectors based on feedback from the EEG decoder. This forms the basis of a dynamic and iterative learning process that is crucial for maximising the EEG biomarker responses.

## 3.2 SYSTEM IMPLEMENTATION

To demonstrate the feasibility of our proposed framework, we conduct a case study using a SSVEP stimulation protocol. Our objective is to elicit the most robust EEG biomarker responses under the SSVEP protocol. We collected EEG data using a dry wearable device (developed by BrainUp research Lab), which has seven channels and can capture neural activity across a wide visual area.

Our system, depicted in Figure 2, includes a pre-trained DCGAN with five layers of specified shapes. To customize the model for our purposes, we create a custom training dataset that featured visual stimuli with variations in brightness, stripe patterns, and checkerboards. The discriminator is constructed as a convolutional neural network, specifically designed to process images with dimensions $64 \times 64 \times 3$ (width $\times$ height $\times$ channels). It incorporates four strided convolutional layers, which sequentially reduce the resolution of the input image. And the generator in the DCGAN framework utilizes a latent vector of length 100, sampled from a standard normal distribution. This vector is processed through a series of transposed convolutional layers to construct an image. More training details are provided in Appendix A.1.

The EEG decoder employs a rapid Fast Fourier Transform (FFT) approach, in conjunction with Signal-to-Noise Ratio (SNR) calculations at the target frequency, both as feature extraction techniques. The FFT feature highlights the maximal response within V1 to the target frequency, effectively identifying the most significant neuronal activation triggered by a specific visual stimulus. Conversely, the SNR feature quantitatively assesses the contrast between the amplitude of the target frequency and the energy of the background oscillations. This metric provides insights into the signal's stability, offering an indication of the robustness and clarity of the neuronal response relative to background noise. Based on these two features, a score is calculated for each image using the

following equation:

$$\text{Score} = \mathcal{N}orm_{fft}\left(FFT(x(t), f_{\text{target}})\right) + \mathcal{N}orm_{snr}\left(SNR(x(t), f_{\text{target}})\right) \quad (3)$$

where $\mathcal{N}orm_{fft}$ is the normalization function of target frequency FFT amplitude, $\mathcal{N}orm_{snr}$ is a normalization function of target frequency SNR. $x(t)$ represents the EEG data trial in the time domain and $f_{\text{target}}$ is the target frequency. The maximum value used for normalization is taken from the values of the eighth iteration.

The latent vector generator algorithm is built based on Equation 1, which $n$ equals two. One latent vector originates from an image with the highest FFT value of the EEG signal, while another latent vector is derived from an image with the highest SNR value of the EEG signal. These two latent vectors are combined to obtain the optimal latent vectors. The optimal latent vector is then used to generate eight individual offspring vectors. In each offspring, a portion of the latent vector is replaced with random numbers drawn from a a zero-centered Gaussian distribution to serve as mutation vectors. The proportion of the vector that is replaced varies among offspring, ranging from $p = 0\%$ to 80%. Additionally, ten individual offspring vectors are generated using the interpolation process between the best FFT and SNR latent vectors. Therefore, the total generated latent vector is 20 for the next iteration. This introduces variability and potentially novel feature representations into the generative process.

## 4 RESULTS

### 4.1 EXPERIMENTAL SETUP

We conducted an experiment involving nine human participants (6 males and 3 females). At the first iteration, a trained generator produced 50 images. Through an image pre-check process (details are described in Appendix A.2), the five most diverse images were selected for further testing using a SSVEP protocol at 4 Hz. The reason for selecting 4 Hz is that it is the frequency of the brain's background noise, which aids significantly in evaluating whether the testing system can effectively evoke the target frequency while suppressing the background noise. Each image was presented for a specified duration (125 milliseconds) per trial. Each trial lasted two seconds. After completing 30 trials, participants were allowed a 30-second period to return to resting states. Then the latent generator used two best images (decided by an EEG decoder) to produce 20 new images for subsequent iterations in the experiment loop. The experimental picture is shown at Figure 3 (a), depicting a typical experimental configuration, in which a participant outfitted with an dry electrode EEG device (BrainUp research Lab developed with 7 recording channels) while engaged with a visual stimulus presented on a computer screen. The electrode location is based on 10-20 system: O1, O2,T5, P3, P4, T6 and Pz, which mainly cover V1 areas. The experimental details are described in Appendix A.3.

### 4.2 VEP BOOSTER RESULTS

Figure 3 (b) presents an analysis of SSVEP responses, comparing the outputs from the VEP Booster with natural responses. The response (EEG biomarker) is calculated by normalizing two feature values (FFT and SNR) and adding them together. The data is calculated by average the results of nine human subjects with standard variations.The data illustrates a distinct trend: responses by the VEP Booster (indicated by red triangles) demonstrate a consistent upward trajectory, suggesting an enhancement in the SSVEP responses. Conversely, the naturally observed responses (represented by gray circles) exhibit substantial variability at lower band. At the last iteration loop, the mean SSVEP response generated by the VEP Booster is 106% higher than the natural ones, with the improvement ranging from 28% to 295%. It is noteworthy that there is a plateau point, where the SSVEP response stabilizes. A slight decline following the plateau may be attributed to neural adaption.

Figure 3 (c) illustrates the heat maps of the VEP Booster and natural experimental results of all trials and iterations, respectively. These panels clearly demonstrate that the SSVEP responses increase with each iteration in the VEP Booster simulations, depicted by an escalating intensity in the heat map. In contrast, the heat maps from natural trials show consistently low random values across the iterations. This distinct pattern suggested the capability of the proposed VEP Booster in successfully generating images that preferentially activate biological visual cortex V1, thereby eliciting increasingly strong EEG biomarker related neuron responses.

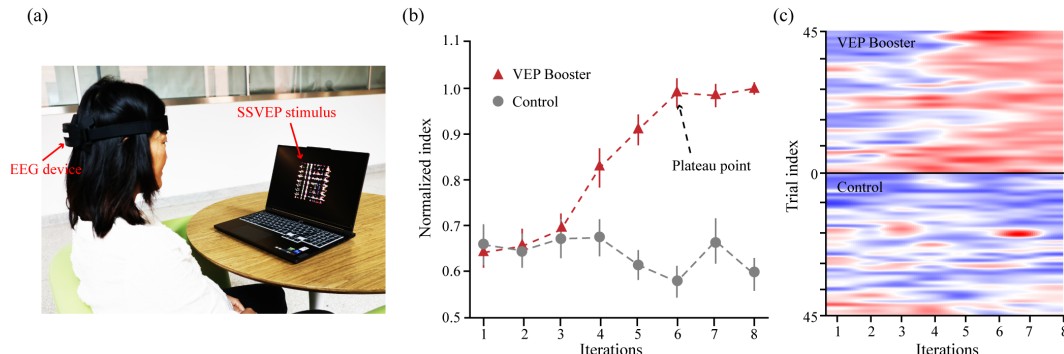

Figure 3: (a) A participant outfitted with an EEG device, actively engaging with a visual stimulus displayed on a computer screen. (b) A comparative analysis of SSVEP (Steady State Visual Evoked Potentials) responses elicited by VEP Booster-generated images versus natural images. The graph presents average values from multiple trials (5 images × 30 trials each) with standard deviations indicated. (c) Heat maps representing the aggregate SSVEP responses to both VEP Booster-generated and natural images from all trial iterations, illustrating the spatial distribution of neural activity.

### 4.3 SSVEP EEG RESULTS

We initiated our investigation by analyzing how the VEP Booster adapts visual stimuli over successive iterations. Figure 4(a) illustrates the iterative changes in the stimulus images, showing how the system refines the visual inputs to better align with the neuronal preferences of the primary visual cortex. More details of VEP generated images are in Appendix A.4.

As depicted in Figure 4(b), there was a significant increase in the EEG biomarker values with each iteration, indicating that the VEP Booster successfully enhances neural responses. Each panel displays EEG traces with the black line representing the mean amplitude of all trials within that respective iteration. The red vertical dashed line indicates the offset of the visual stimulus. Accompanying each EEG trace, brain topography maps provide a spatial representation of neural activity, emphasizing regions of significant activation. The data reveals a progressive enhancement in signal quality as indicated by the signal-to-noise ratio (SNR) values across the iterations. Starting from an SNR of 1.51 in the first iteration, there is a clear trend of increasing SNR, reaching up to 2.75 in the eighth iteration. Moreover, the amplitude of the EEG responses also shows a significant increase, from 123 $\mu$V in the first iteration to 503 $\mu$V in the eighth, which supports the SNR findings.

### 4.4 EEG BIOMARKER ACROSS NINE SUBJECTS

The efficacy of the Visual Evoked Potential (VEP) Booster was systematically evaluated across nine subjects to verify its generalizability and robustness in enhancing EEG biomarkers. As depicted in Figure 5(a), violin plots illustrate a notable increase in both amplitude and SNR from the first to the last iteration across all participants. For instance, Subject 1 exhibited an increase in amplitude from 111.1 to 255.2 and in SNR from 1.32 to 2.66, while Subject 9 showed enhancements from 60.98 to 108.74 in amplitude and from 1.81 to 1.91 in SNR. This uniform elevation is particularly evident in the amplitude measures, where the median values shift markedly upwards, accompanied by a broadening in the distribution of the last iteration values, indicating a pronounced and consistent enhancement in response to the VEP Booster. Furthermore, Figure 5(b) reinforces these findings by providing a detailed analysis per subject, demonstrating not only significant improvements in amplitude and SNR values but also consistency across trials. The overlay of improvement bars on the line plots quantifies these enhancements, with the most significant observed increase in amplitude being for Subject 4, from 245.3 to 503.8, and the greatest improvement in SNR for Subject 6, from 2.32 to 3.36. The T-test results indicated in the charts (T = -3.97, p = 0.004 and T = -4.05, p = 0.004), significantly below the conventional significance level of 0.05, strongly support rejecting the null hypothesis that there is no difference between the two iterations.

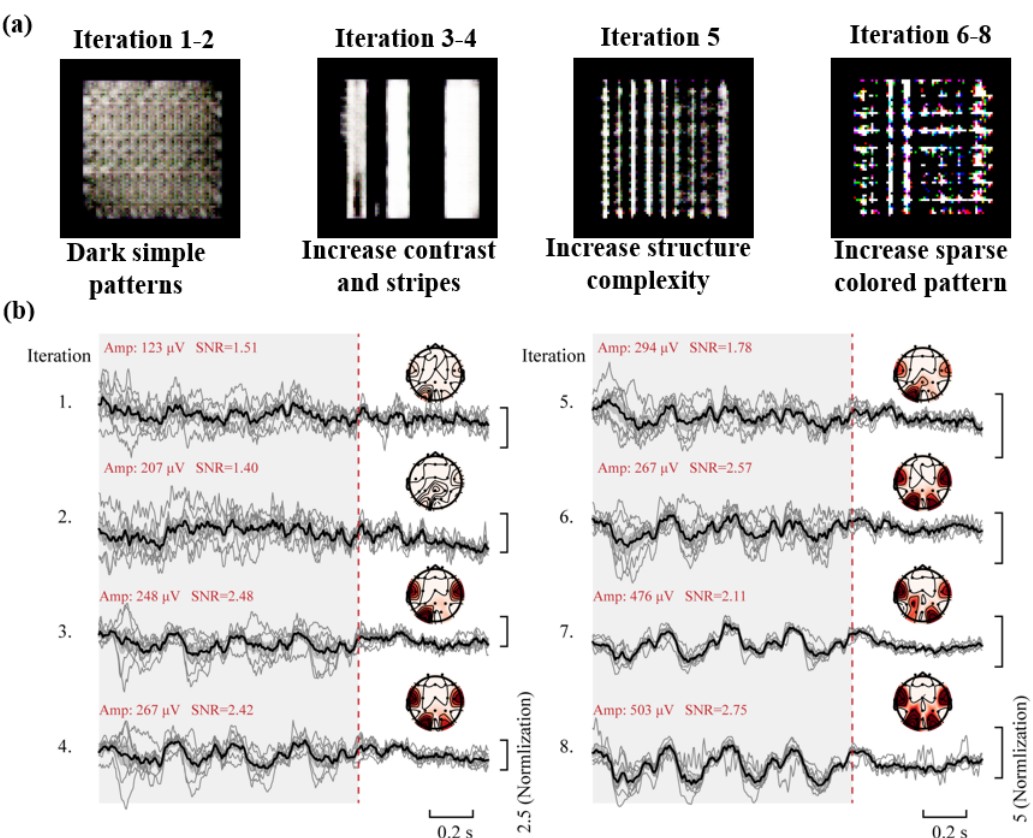

Figure 4: (a) A brief summary of how image evolves; (b) Sequential EEG data and brain topography from eight iterations for a participant. Each panel displays the EEG responses for a single iteration, with the black line representing the average amplitude of all trials within that iteration. The corresponding brain topography maps illustrate the distribution of neural activities. The amplitude (Amp) and signal-to-noise ratio (SNR) for each iteration are noted, demonstrating a progression in response clarity and strength across the sessions.

### 4.5 CROSS-TIME CHANGES IN EEG BIOMARKERS AND KERNEL DENSITY DISTRIBUTION

Three subjects were enrolled to participate in cross-time EEG biomarker change experiments, which were conducted over a 12-hour period, commencing at 10 AM and concluding at 10 PM. During this period, the total experiment number is five, each subject participated in two sets of steady-state visual evoked potential (SSVEP) experiments, with a two-hour interval between each experiment. The initiation times for the experiments were randomized. For each experiment set, images from the initial (control) and final (VEP booster-enhanced) iterations were selected for analysis. Each image was presented in thirty trials, with each trial consisting of a one-second stimulation period followed by a one-second resting period.

Over the time course of 10 hours, by average the three human subject results, Figure 6(a) displays the EEG biomarker values for the VEP booster images consistently exceeded those of the natural images. The statistically significant T-test results, T = -3.97, p = 0.01, augment the evidence supporting the superiority of the VEP booster images. Additionally, despite variations in brain state over time affecting both sets of image indices. The VEP booster images indices exhibited less variations (0.065) compared to the natural images' indices (0.0691). This demonstrates that the VEP booster images produce biomarkers that are stable and less affected by background brain oscillations and other factors.

Notably, between 4-hr and 6-hr here was a significant decline with natural image-based EEG indices dropping by 20% (from 0.92 to 0.74) and VEP booster-based EEG indices biomarkers only declining

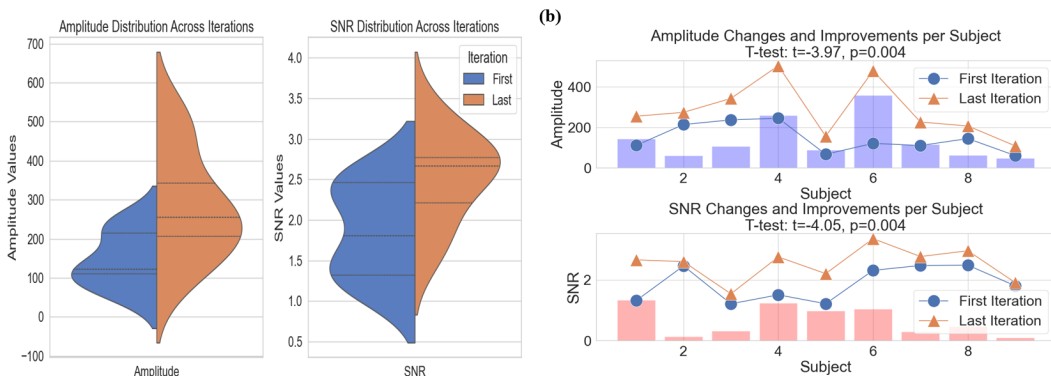

Figure 5: (a) Violin plots demonstrating the distributions of amplitude and SNR values for all participants before and after using the VEP Booster, showing marked improvements.(b) Detailed improvements per subject illustrated through line and bar plots, highlighting consistent and significant increases in amplitude and SNR across iterations.

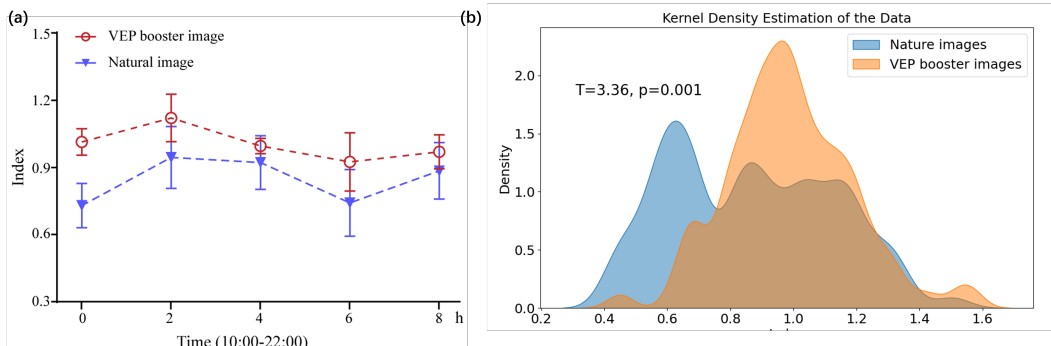

Figure 6: (a) SSVEP EEG biomarker indices recorded from 10:00 to 22:00, showing the responses to VEP booster images (red) and natural images (blue). (b)Kernel density estimates of the data reveal distinct distribution patterns for biomarker indices.

by 7% (from 0.99 to 0.92). One of the reasons is due to changes in the brain's background oscillations within this time period. Since the VEP booster image can induces stronger EEG biomarkers (with SSVEP natural images, the SSVEP biomarker was 123uV with an SNR of 1.51, and with VEP booster images, the SSVEP biomarker increased to 503uV with an SNR of 2.75), it is less affected by background oscillations.

Figure 6(b) depicts the kernel density distribution for VEP booster images (depicted in blue) showed a prominent peak around the index of 1, whereas the distribution for natural images (depicted in orange) was broader with a peak around 0.6. These results indicated that EEG signals from VEP booster images had higher indices concentrated in a narrower range with lower variability in comparison to those from natural images. These results indicated that VEP booster has improved stationarity and less variability in EEG signals.

### 4.6 INTRA-SEGMENT TRAINING AND VALIDATION

To further substantiate the efficacy of the VEP Booster, we conducted a binary classification task utilizing Steady-State Visual Evoked Potentials (SSVEP). The objective was to determine whether human subjects were under SSVEP stimulation. For this task, we selected multiple neural network models. Given that our study spanned five distinct time periods, we compiled a dataset comprising 4500 trials (more details are in Appendix A.5. By training and testing the model on 80% and 20% splits within the all time periods data, the strategy focuses on capturing and adapting to the nuances

Table 1: Comparison of different neural network accuracy across multiple time points

| Time periods | VEP Booster Condition | CCNN (%) | EEGNet (%) | SSVEPNet (%) |
|---|---|---|---|---|
| 10am-12am | Without | $88.89 \pm 9.94$ | $89.44 \pm 9.84$ | $87.78 \pm 13.90$ |
| | With | $87.22 \pm 11.33$ | $85.56 \pm 12.35$ | $91.11 \pm 8.85$ |
| 12am-2pm | Without | $88.33 \pm 10.00$ | $91.67 \pm 8.61$ | $88.33 \pm 5.93$ |
| | With | $94.44 \pm 7.86$ | $93.34 \pm 6.24$ | $88.89 \pm 8.96$ |
| 2pm-4pm | Without | $78.33 \pm 9.03$ | $83.89 \pm 11.97$ | $80.56 \pm 12.42$ |
| | With | $82.22 \pm 10.92$ | $87.22 \pm 12.50$ | $88.89 \pm 8.96$ |
| 4pm-6pm | Without | $91.11 \pm 8.32$ | $90.56 \pm 10.48$ | $88.89 \pm 11.65$ |
| | With | $93.33 \pm 8.71$ | $95.00 \pm 7.33$ | $93.33 \pm 8.16$ |
| 6pm-8(10)pm | Without | $91.67 \pm 9.13$ | $90.55 \pm 8.54$ | $91.67 \pm 8.61$ |
| | With | $95.56 \pm 5.15$ | $96.11 \pm 6.71$ | $88.89 \pm 12.04$ |
| 10am-10pm | Without | $86.44 \pm 0.6$ | $88.15 \pm 1$ | $87.86 \pm 1$ |
| | With | $91 \pm 0.7$ | $91.24 \pm 0.6$ | $92.12 \pm 0.9$ |

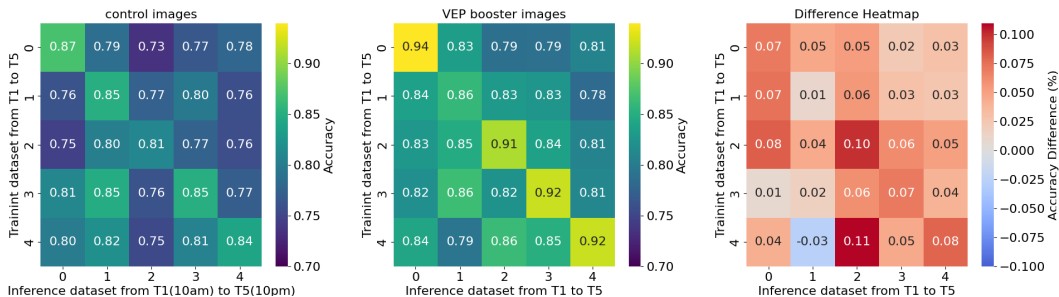

Figure 7: Comparison of CCNN classification accuracy across different time periods with and without the VEP Booster.

within a single period. We conducted five-fold cross-validation to validate the models' performance. The results, showing mean and standard deviation of accuracy, are detailed in the table below.

The practical utility of the VEP Booster in enhancing neural network performance is clearly evidenced through a systematic comparison of classification accuracies across multiple time points, as summarized in Table 1. Notably, for the SSVEPNet, the mean accuracy across all the time periods are above 88%, while without VEP booster, the accuracy range from 81% to 91% with a considerable variations. This trend persisted throughout the experiment, culminating in the 10am-10pm overall comparison where the introduction of the VEP Booster led to an increase in accuracies: CCNN from 86.44% to 91%, EEGNet from 88.15% to 91.24%, and SSVEPNet from 87.86% to 92.12%.

### 4.7 INTER-SEGMENT TRAINING AND VALIDATION

Also, we tests the model's generalization capability with only a single time period data. The structured testing involved training the model on a dataset from a single time period (900 samples) and validating it across subsequent periods. The results, Figure 7depicted in the provided heatmap comparisons, clearly highlight the VEP Booster's effectiveness. When employing the CCNN with VEP Booster, the difference Heatmap highlights the improvement in accuracy with increases ranging from 1% to 11% in accuracy points. At T3 time period case, the overall accuracy across all time perods is above 81%, while at control set the accuracy varies between 75% and 81%. In addition, the other two neual networks results are shown at Appendix A.6.

### 4.8 VEP BOOSTER CONVERGENCE AND FEATURE REPRESENTATION

We employed Kernel Principal Component Analysis (KPCA) as our dimensionality reduction technique to transform the latent vectors from a high-dimensional space (100 dimensions) to a more manageable three-dimensional space. Therefore, we explore the dynamics of a VEP Booster within

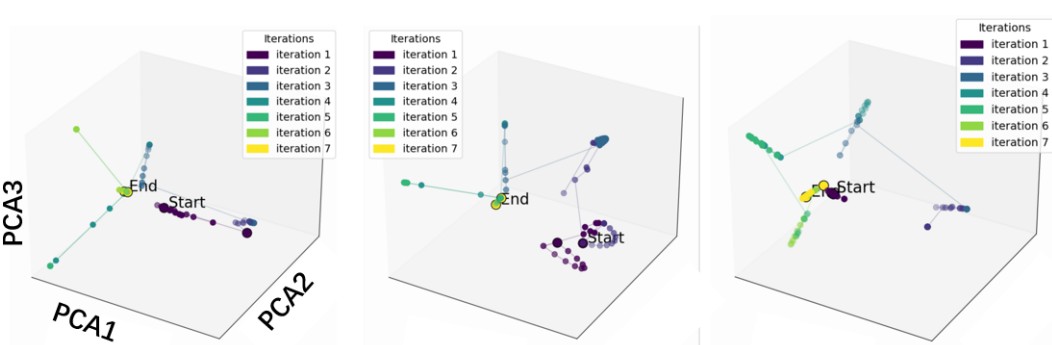

Figure 8: Trajectories of three human subject latent vectors in a 3D latent space visualized across different iterations. The visualization illustrates the convergence patterns from initial ('Start') to final ('End') positions for selected groups (iteration 1 and 7) over multiple iterations.

the framework of a DCGAN, particularly focusing on the trajectory of latent vectors in a high-dimensional space. First, the trajectories in the latent space provide compelling evidence of the system's ability to converge towards stable solutions. By examining the start and end points of these trajectories, as highlighted in the accompanying visualizations, we observe that the network systematically refines its internal representations towards more compact areas of the latent space. This convergence is not only rapid but also consistent across different initialization conditions, underlining the robustness of our model in finding optimal solutions. Second, the trajectory paths provide insight into the representation of information by DCGANs and offer a reflection of how the biological visual system encodes such information. It help us to understand and develop the bio-inspired neural network learns to capture and replicate key aspects of biological visual neural network information representation.

## 5 CONCLUSION

In this paper, we introduced the VEP Booster, a novel closed-loop AI framework designed to optimize reliable and stable EEG biomarkers. Utilizing a SSVEP protocol, our findings demonstrate that this system can robustly evoke individual biomarkers compared to natural states for all individuals. In the context of single-trial Steady-State Visual Evoked Potential (SSVEP) binary classification experiments, both intra- and inter-temporal analyses with the Visual Evoked Potential (VEP) Booster yielded statistically significant enhancements.

Also, our results have led us to identify several key factors that are critical for the effective operation of the system. First, the optimization of the DCGAN and EEG decoder must be synchronized to ensure that the decoder can detect subtle differences in generated images. Second, it is essential to balance the GAN loss and the EEG decoder capabilities to prevent either aspect from dominating the learning process, which could result in underfitting one side of the model.

Regarding board applications, the VEP booster's core capability is to generate customized visual stimuli based on real-time EEG feedback, which is fundamental not only for SSVEP but also for other types of visual protocols such as flash VEPs (fVEPs) and Event-related Potentials (ERPs), which enabling a broader application scope of visual stimulation protocols. This system's performance highlights its utility in providing tailored therapeutic interventions, promising a shift towards more personalized and effective treatment strategies in neurological care.

## 6 REPRODUCIBILITY STATEMENT

We will release the code and usage instructions to the public upon acceptance.

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

# A    APPENDIX

## A.1    IMPLEMENTATION DETAILS

The dataset was programmatically synthesized using a Python script and consists of images standardized to a resolution of 1024x1024 pixels against a black background. This dataset is categorized into three distinct groups, with each group containing approximately 60 images. The first category encapsulates a gradational luminance shift from black to white. The second category comprises images with uniformly spaced black and white striped patterns. The third category features images with a progressively dense checkerboard grid.

The discriminator is constructed as a convolutional neural network, specifically designed to process images with dimensions $64 \times 64 \times 3$ (width $\times$ height $\times$ channels). It incorporates four strided convolutional layers, which sequentially reduce the resolution of the input image. The layers are configured as follows:

- **Layer 1:** Convolutional layer, accepting an input of $64 \times 64 \times 3$, producing an output of $32 \times 32 \times 64$.

- **Layer 2:** Strided convolutional layer, outputting $16 \times 16 \times 128$.

- **Layer 3:** Strided convolutional layer, further reducing to $8 \times 8 \times 256$.

- **Layer 4:** Strided convolutional layer, with a final output of $4 \times 4 \times 512$.

Each convolutional layer, except the first, is followed by batch normalization and a LeakyReLU activation function to improve training stability and introduce non-linearity. The output of the final layer is passed through a sigmoid activation function, yielding a single scalar value. This scalar represents the probability that the input image is classified as real.

The generator in the Deep DCGAN framework utilizes a latent vector of length 100, sampled from a standard normal distribution. This vector is processed through a series of transposed convolutional layers to construct an image:

- **Layer 1:** Transposed convolutional layer, converts the latent vector into a $4 \times 4 \times 1024$ feature map.

- **Layer 2:** Transposed convolutional layer, upscales to $8 \times 8 \times 512$.

- **Layer 3:** Transposed convolutional layer, further enlarges to $16 \times 16 \times 256$.

- **Layer 4:** Transposed convolutional layer, increases resolution to $32 \times 32 \times 128$.

- **Final Layer:** Transposed convolutional layer, producing a full-resolution image of $64 \times 64 \times 3$.

Batch normalization and ReLU activations are used in each layer, except in the final layer, where a tanh activation function is used to normalize the image pixels between -1 and 1. This architecture allows the generator to transform a simple distribution into complex data structures that mimic the training images. All experiments were conducted on a computer running Linux Ubuntu 22.04 system, Python version 3.8 and pyTorch 1.10 . The hardware components are as follows: CPU: Intel Xeon 6230 2.1GHz MEM: DDR4 1 TB DISK: 8 TB GPU: Quadro RTX 6000, 24 GB.

## A.2 Image Pre-check Process

The image pre-check process extract a suite of five distinct features: standard deviation of pixel values, edge count determined through the Sobel operator, energy of the high-frequency component from Haar wavelet decomposition, mean frequency from the Fourier transform, and skewness of the pixel intensity histogram. These features encapsulate various aspects of the texture of the image, the information about the edges, the frequency content, and the distribution characteristics. Based on these feature values, it computes the total pairwise Euclidean distances between the feature vectors of the selected images. This metric serves as a quantitative measure of diversity, with the assumption that a higher sum of distances indicates greater dissimilarity among images.

## A.3 EEG Recording Experimental and Visual Stimulation Protocol Setup

The wearable EEG acquisition system comprised nine channels, including seven for EEG acquisition, one for reference, and one for bias. The microcontroller unit (MCU) served as the control unit, interfacing with the analog-to-digital converter (ADC) via serial input/output and communicating with the workstation using Bluetooth 5.0 protocol. Raw EEG signals were filtered using a 50 Hz band stop filter and sampled at 250 Hz. The amplitude and amplitude of the steady-state visually evoked potential (SSVEP) at the target frequency of 4 Hz, along with the signal-to-noise ratio (SNR), were analyzed using Fast Fourier Transform (FFT). For each stimulation frequency, the EEG data from each trial were segmented into equal epochs starting from the initial timestamp.

Data collection involved nine healthy subjects (aged 22-36 years; 3 females and 6 males). The recordings were conducted in a quiet, dimly lit environment, isolated from known sources of electrical interference. Informed consent was obtained from all participants and the experimental procedures were approved by the Ethics Committee.

**The generated image in eight iterations**

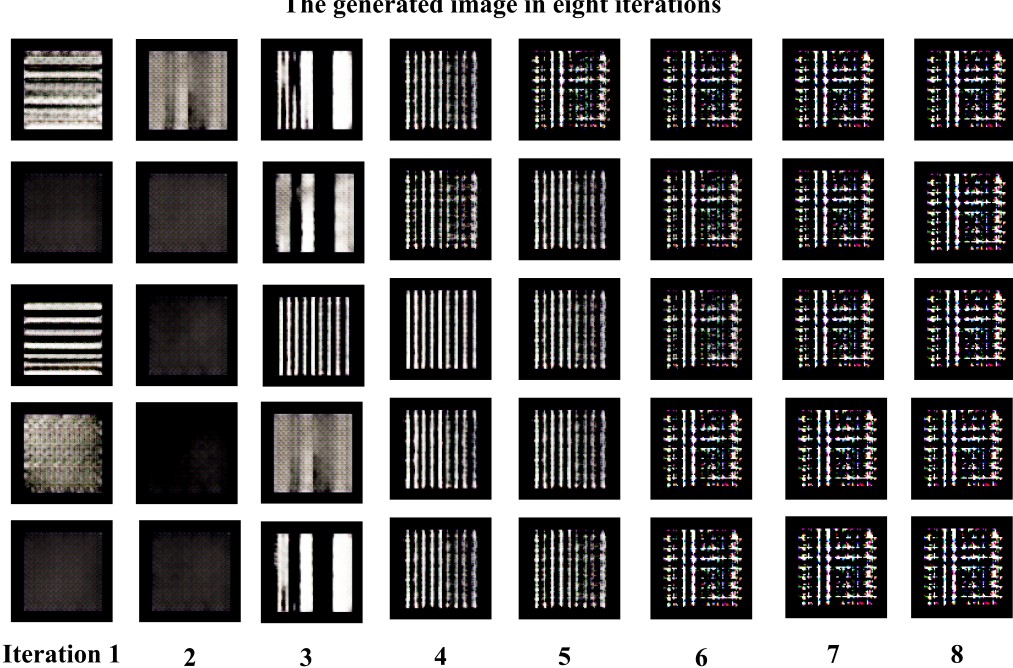

Iteration 1    2    3    4    5    6    7    8

Figure A.1: The VEP generated images across eight iterations of a participant.

The stimulus images consisted of a square that underwent a on-off modulation of temporal contrast 100%, displayed in the center of an LCD screen with a resolution of $2560 \times 1600$ pixels, positioned 60 cm from the subjects' eyes. The stimuli had a visual angle of approximately $6° \times 6°$. The stimulus was presented at a flicker frequency of 4 Hz (125 ms on and 125 ms off), which was used to detect the response at this specific frequency. Each iteration consisted of 30 trials with a 30-ms interval between the trials, with up to 8 iterations conducted.

The normalized index and amplitude of SSVEP were analyzed using the Python toolbox. The data was presented by mean±s.t.d. The shielding area in the mean response indicates the 95% confidence interval.

### A.4    THE VEP BOOSTER GENERATED IMAGES

For a single human participant, the VEP booster generated images of eight iterations are in Figure A.1.

### A.5    CROSS TIME TRAINING DATASET

This dataset comprises electroencephalography (EEG) data collected using a wireless dry-electrode system with a sampling rate of 250 Hz. It includes experimental data from three subjects, each participating in experiments across five different time periods (from 10:00 AM to 10:00 PM at two-hour intervals). In each time period, subjects completed two sets of experiments: a control group and a Visual Evoked Potential (VEP) group. Due to wireless transmission, some data points may be missing, resulting in approximately 500 data points per trial.

In this study, electroencephalography (EEG) data were collected from three subjects who participated in experiments across five different time periods, scheduled every two hours from 10:00AM to 10:00PM. The experimental design employed a Steady-State Visual Evoked Potential (SSVEP) paradigm using five natural images as stimuli. Each image was presented in 30 trials lasting 2 seconds each, resulting in 150 trials for the control group per time period. In addition, a VEP group was included, where subjects were shown five images generated by a VEP enhancer, also in 30 trials per

Figure A.2: Comparison of classification accuracy across different time periods with and without the VEP Booster (SVEPNet, EEGNet and CCNN).

image with each trial lasting 2 seconds (comprising 1 second of SSVEP stimulation and 1 second of resting state), yielding another 150 trials per time period. This led to a total of 300 trials per time period per subject, amounting to 1,500 trials per subject over all time periods and 4,500 trials across all subjects. The EEG data were stored in .npy (NumPy array) files with dimensions (30, 8, 500), where 30 represents the number of trials per time period, 8 denotes the number of channels (the first seven are EEG data channels and the eighth is a marker channel without data), and 500 indicates the number of data points per trial, which may slightly vary due to missing data.

## A.6  FULL RESULTS ON INTER-SEGMENT TRAINING AND VALIDATION

The full results are shown at Figure A.2.

