# OpenReview forum: "The VEP Booster: A Closed-Loop AI System for Visual EEG Biomarker Auto-generation"
_ICLR.cc/2025/Conference — ICLR 2025 Conference Withdrawn Submission_

### Official Review · Reviewer_BN5d · 2024-11-02

**Soundness:** 2
**Presentation:** 3
**Contribution:** 2
**Rating:** 3
**Confidence:** 4

**Summary:**

The authors propose the Visual Evoked Potential
Booster (VEP Booster), a novel closed-loop machine learning pipeline designed
to produce reliable and stable EEG biomarkers under visual stimulation
protocols. The system leverages Deep Convolutional Generative Adversarial Network
(DCGAN) to refine stimulus images based on real-time feedback from human
EEG signals. As a result, it generates visual stimuli that are specifically tailored to the primary visual cortex of the participant.

The authors assessed the efficacy of this system through the implementation of steady-state visual
evoked potential (SSVEP) protocols in nine human subjects. The results show statistically significant enhancements when employing the VEP Booster.

**Strengths:**

The paper proposes an interesting new closed-loop BMI system for VEPs. The numerical results are promising, and the methodology is generally applicable.

**Weaknesses:**

On the one hand, the technical contributions are limited, and also only briefly described. Therefore, I fear that this submission will only be of interest to a small group of researchers in the ICLR community. On the other hand, the neuroscientific aspects are also only superficially described. How are the stimuli precisely designed? Why do we need a GAN for this purpose? I believe that the stimuli are quite simple, and they could be rendered by a simple scripts instead of a GAN. What can be learn from the optimized patterns in Fig 4 in terms of biophysics? It is also not clear how this paradigm could be used in practical applications. I recommend providing specific examples of potential practical applications for the proposed system, or to elaborate on the challenges that might need to be addressed for real-world implementation.

In summary, the paper may only have limited appeal to the methods-oriented audience on the one hand and neuroscience community on the other hand, since the methodology and interpretation of the results is too superficial.

**Questions:**

How are the stimuli precisely designed?

Why do we need a GAN for this purpose? I believe that the stimuli are quite simple, and they could be rendered by a simple scripts instead of a GAN.

What can be learn from the optimized patterns in Fig 4 in terms of biophysics?

---

### Official Review · Reviewer_p4ZU · 2024-11-03

**Soundness:** 2
**Presentation:** 2
**Contribution:** 4
**Rating:** 6
**Confidence:** 4

**Summary:**

This paper presents the Visual Evoked Potential Booster (VEP Booster), a closed-loop AI system aimed at generating reliable and stable EEG biomarkers through visual stimulation protocols. The VEP Booster uses a GAN to refine visual stimuli based on real-time EEG feedback, targeting neurons in the primary visual cortex to optimize EEG responses. Experimental results demonstrate significant improvements in EEG biomarkers, highlighting the system's potential for clinical and technological applications.

**Strengths:**

The paper proposes a compelling approach by leveraging a closed-loop system combined with generative models to create tailored visual stimuli for EEG. This method effectively reduces individual differences and improves the quality of EEG recordings, which is essential for broader applications in brain-machine interfaces. In this updated version, the authors have added substantial experiments and analyses, reinforcing the demonstrated effectiveness of the approach.

**Weaknesses:**

1. The generated stimuli are overly simple, as the model primarily produces variations in brightness, stripe patterns, and checkerboards. These could likely be constructed with a simpler model with fewer parameters, rather than a complex DCGAN, which would reduce optimization challenges. Additionally, if a complex model like a DCGAN is employed, it would be feasible to apply this framework with natural images, which would be significantly more flexible and practical than the current stimuli.
2. Typos: The graphic in Figure 5a appears stretched； Figure A.2, subtitle for the 3rd row, "CCNN" should be corrected.

**Questions:**

1. The paper frequently contrasts "natural images" with "VEP booster images," but does not clearly define "natural images." It would be helpful if the authors could clarify how these images are constructed. Additionally, the term "natural images" may be misleading, as it could be interpreted as real-world scenes, which does not seem to be the intent here. If so, a simpler description like "control image" may be more appropriate.

2. Why not use a neural network classifier as the EEG decoder directly? Given that neural networks are fully differentiable, defining a target class might allow the current framework to operate in a similar way.

---

### Official Review · Reviewer_qdND · 2024-11-03

**Soundness:** 1
**Presentation:** 2
**Contribution:** 1
**Rating:** 3
**Confidence:** 4

**Summary:**

The authors present a novel approach to improve the salience of  EEG biomarkers during visual stimulation under the Steady State Visual Evoked Potentials methodology. They utilize a closed-loop paradigm where, firstly, the DCGAN generates images. These images are being chosen as stimuli for a participant. A participant views the generated stimuli and their brain activity is being recorded. Then, two latent vectors with the highest scores are chosen and mutated to create a range of new latent vectors which closes the loop based on measuring the quality of the elicited by the new image EEG signal. The stimuli are being presented using the SSVEP protocol to cause oscillatory entrainment in the visual cortex.  Experiments are done for nine subjects. Authors show the statistically significant increases in metrics associated with the quality of biomarkers (SNR at the corresponding frequency) during this approach vs using natural images as visual stimuli.

**Strengths:**

1. The authors present a novel closed-loop neural network-based approach that designed to iteratively improve EEG-based biomarkers quality and strength
2. The authors collected a new dataset and provided a detailed explanation of the experiment and the main ideas of the approach with well-made and useful graphics
3. Globally, reverse engineering human’s sensory systems using closed-loop paradigms based on the non-invasive brain activity recordings is a promising state-of-the-art trend of the modern neuroscience

**Weaknesses:**

1. **Lack of comparison with other methods**. First of all, the authors provide comparisons of NN-modified Gabor-like structures and measure their ability to elicit the frequency entrainment, comparing it to the  natural images. SSVEP paradigm used by the authors is well known and there have been numerous studies dating 10+ years ago and the recent ones aimed to optimize the spatial and temporal parameters of the visual stimuli used in the SSVEP paradigm in order to improve the detectability.  While the authors provide numerous references to other approaches in Section 2 including traditional signal processing techniques, machine learning (and neural network)-based techniques and Spiking Neural Networks, they do not compare their approach with at least some of the most closely related ones.
2. **Small dataset.** The proposed dataset includes 9 subjects and 4500 trials. This is generally a pretty small amount of data. Therefore, it is not clear how well the approach scales and generalizes across subjects. Also, one of the experiments authors do is to train a neural network to classify if a subject was under SSVEP stimulation or not. They use a traditional 80-20 split which leaves only 900 samples for testing. It is also possible that neural networks’ quality was affected by the lack of data during training. At the same time the modern SSVEP paradigms are capable of designing neurointerfaces with 16-32 command where each command is encoded with a unique visual pattern flickering at a given rate. Pushing the limits of such SSVEP solutions with the proposed approach and comparing it against the earlier methods would significantly increase the impact of the study.
3. **Potential model-specific ML issues. Mode collapse.** GANs are famously prone to a mode collapse where a model starts to generate a really small variety of outputs. Considering that authors were not only using GAN, but also finetuning it, it is important to address if a GAN actually learned to produce images well and was it affected by a mode collapse. This seems especially important to address since authors talk about convergence of the system to the same latent vectors (as seen in Figure 8 and Figure A.1). As it could be a sign of a mode collapse affecting the closed-loop experiment. Also, are the authors sure that they truly need large NN architectures to solve their task?
4. **Reproducibility.** It would be very useful if authors could provide more details about how their model was trained: optimizers, loss functions, etc. Also, authors utilize a dataset with 9 people, but in the Introduction (lines 98-99) they say they will open-source data only from 3 subjects. This raises the reproducibility concerns. Especially, considering weakness (1): authors did not try other models on their data and other authors will not be able to train their models for direct comparison with this approach without having access to the same full dataset.
5. **Incorrect statements**. While I understand that the authors are primarily people working on the ML, I still think the text has to be monitored for incorrect and very naive statements, For example such as “The reason for selecting 4 Hz is that it is the frequency of the brain’s background noise, which aids significantly in evaluating whether the testing system can effectively evoke the target frequency while suppressing the background noise.” - It is well known both neuroscience and EEG signal processing community that the brain’s background noise has PDF of the form  P(f) ~ 1/f^alpha and is not centered around 4 Hz.  4 Hz fall in the range of so called theta-band known to be associated with memory related processes.

**Questions:**

1. Could the authors compare their approach with some of the models listed in Section “2 Related work”?
2. Could the authors clarify the training process for DCGAN and for CCNN, EEGNet and SSVEPNet? Were you training subject-specific models or subject-independent (using data from all subjects at the same time)?
3. Could the authors compare the performance of the NN-based models to a more classical and well accepted in the SSVEP community CCA approach, e.g. https://pubmed.ncbi.nlm.nih.gov/25081427/
4. Why do the authors compare their approach against stimulation with natural images and not with the ones similarly structured (nearly periodic spatially) to your GAN outputs?
5. Why did the authors decide to use GAN and not any more modern image generator (e.g. Diffusion Models)?
6, What was the distribution of the data used for binary classification? It is important as classification accuracy is prone to class imbalance.

---

### Official Review · Reviewer_sGi5 · 2024-11-03

**Soundness:** 2
**Presentation:** 1
**Contribution:** 2
**Rating:** 3
**Confidence:** 4

**Summary:**

The paper describes a closed-loop pipeline to maximize the discriminability of visually evoked potentials measured with EEG brain data. Specifically, a DCGAN is used to generate SSVEP stimulus images, and the distribution of the input latent vectors to its Generator are iteratively modified such that measures of SSVEP discriminability are increased from trial to trial. Results on 9 subjects suggest the proposed approach can indeed improve SSVEP-derived features significantly and at different times during the day.

**Strengths:**

* Originality: The use of a DCGAN-based closed-loop system for optimizing visual stimuli appears novel.
* Clarity: Though there is missing important information (see Q1-4), the overall approach and experiments are mostly clear.
* Significance: The proposed approach paves the way to dynamically improving stimulation parameters in BCI tasks.

**Weaknesses:**

1. There are multiple unsubstantiated claims about EEG: the intro talks about “intrinsic preferences of specific neurons” however there is no way to identify those with EEG. This also comes up at line 211 (“maximal response within V1”), line 241 (“4 Hz [...] is the frequency of the brain’s background noise”) and line 269 (“generating images that preferentially activate biological visual cortex V1”).
2. A lot of technical details require clarification including the way latent vectors are generated, modified and sampled to create trial images (Q1-3), and how the two main metrics used in the study are computed (Q4). Results in Figure 3 are not clearly labeled/explained.
3. While I agree that the data that was collected in this study would be useful to share, listing it as one of the three core contributions of the paper is disputable as it has not been shared yet.
4. The text needs to be proofread.

**Questions:**

1. Are the 50 original generated images the same for all 9 participants, or were different random seeds used for each participant? Am I understanding correctly that only 30 of these are then used in the first iteration (line 732) therefore 20 of those are automatically discarded?
2. My understanding is that after the first iteration, the best latent according to the FFT metric and the best latent according to the SNR metric are averaged to obtain the “optimal latent vector”. An additional 8 “offspring vectors” are obtained by randomly replacing some dimensions by Gaussian noise. Another additional 10 “offspring vectors” are generated by interpolating between the two original vectors. First, how are the 10 interpolated vectors obtained, and does this set overlap with the “optimal latent vector” which is also an interpolation? Second, this gives 19 vectors, but the text (line 231) says 20 - isn’t one missing? Finally, the text says 30 trials were shown per iteration (line 732), but it is not clear how these 20 (or 19) images are distributed among the 30 trials.
3. What is $\alpha$ in Equation 2?
4. There are many details that I don’t understand in Equation 3:

    a. The FFT outputs complex values, but it is not clear whether the phase information is used. Also, what parameters were used for the FFT (e.g. window size, padding, etc.)?

    b. What does the SNR operation do, i.e. what is the exact computation it performs?

    c. What is the second input to the FFT and SNR functions? If it relates to the indexing of the FFT or SNR outputs at frequency $f_\{target}$ it might be clearer to indicate it with a subscript. I believe this argument might be mistakenly included in the FFT and SNR functions, but actually belongs to the Norm functions?

    d. What do the normalization operations do, i.e. are they just dividing each bin by a single value? Why is the maximum value taken from the eighth iteration specifically? What value is used before the eight iteration then?

5. Figure 3: From panel (b), can you clarify how the “106% higher” figure was computed? Also, are these results for a single subject, or are these averaged across all 9 subjects? In panel (c), I don’t understand how this represents the *spatial* distribution of neural activity - is the y axis actually EEG channels (instead of trial indices)? Also, what does the colormap represent?
6. Figure A.1: Can you show the final stimulus pattern obtained for each one of the 9 subjects? Did they converge towards similar patterns? Also, what are the 5 rows in this figure? Finally, why do the images stop changing around iteration 6? If the offspring vectors continue to be generated by randomly replacing some dimensions by random noise, shouldn’t they keep changing?
7. Suppl A.2: “mean frequency of the Fourier transform” → do you mean expected value?
8. Suppl A.3: what is the “Python toolbox” (line 734)?
9. Figure 8 is hard to interpret, focusing on the first two principal components might be clearer (as well as using a nonlinear projection such as UMAP).
10. Others: Suppl A.3: “30-ms interval” → “30-s interval”? In Figure 6 the xlabel is cropped.

---

### Author Response · Authors · 2024-11-25

We greatly appreciate the comments and suggestions from the reviewers, which have been invaluable in enhancing the quality of this work. We are also very grateful for the positive feedback from Reviewer p4ZU.

Based on the comprehensive feedback from the reviewers and discussions within our team, we have decided to withdraw our current submission due to concerns about the completeness of this work.

Finally, we thank you for your input and apologize for not providing timely and effective feedback within the prescribed timeframe.

---

### Note · Authors · 2024-11-25

I have read and agree with the venue's withdrawal policy on behalf of myself and my co-authors.